# Body Image Dissatisfaction Is Increased in Inflammatory Bowel Disease Compared to Healthy Matched Controls but Not Diseased Controls: A Case-Control Study from New Zealand

**DOI:** 10.3390/nu17010015

**Published:** 2024-12-24

**Authors:** Stephen Inns, Heidi Su, Amanda Chen, Crispin Ovenden, Joy Alcantara, Priyanka Lilic, Helen Myint

**Affiliations:** 1Gastroenterology Department, Health New Zealand Te Whatu Ora, Lower Hutt 5040, New Zealand; heidi.su@otago.ac.nz (H.S.); amanda.chen@ccdhb.org.nz (A.C.); cove002@gmail.com (C.O.); joy.alcantara@huttvalleydhb.org.nz (J.A.); priyankalilic@gmail.com (P.L.); helenmyint@gmail.com (H.M.); 2Department of Medicine, University of Otago, Wellington 6242, New Zealand

**Keywords:** inflammatory bowel disease, ulcerative colitis, Crohn’s disease, body image, type 1 diabetes

## Abstract

**Background**: Body image dissatisfaction is elevated in inflammatory bowel disease (IBD) as well as other chronic diseases. The aim of this study was to determine if the higher rate of body image dissatisfaction in IBD is specific to IBD or characteristic of chronic disease in general by comparing body image dissatisfaction in IBD patients with age- and gender-matched healthy individuals and those with type 1 diabetes mellitus (T1DM). **Methods**: In this New Zealand-based case-control study conducted in a secondary care hospital, consecutive IBD patients aged 16 years and older were matched 1:1 with healthy individuals and T1DM patients based on age and gender. However, availability of controls resulted in a slightly different number of pairs for each comparison between groups. Demographics were documented, and participants completed the Body Image Disturbance Questionnaire (BIDQ), RAND SF-36 Quality of Life measures, and Hospital Anxiety and Depression Scale (HADS). **Results**: Forty-five matched pairs compared IBD patients to healthy controls, while 38 compared IBD patients to T1DM patients. BIDQ scores were higher for IBD patients than healthy controls (2.05 vs. 1.58, *p* = 0.004) but not significantly different from T1DM patients (2.03 vs. 1.72, *p* = 0.09). No differences were seen in BMI, smoking, or relationship status across groups. IBD patients had higher depression scores than controls (mean 6.51 vs. 3.87, *p* = 0.002) but similar anxiety scores (5.51 vs. 4.89, *p* = 0.258). A 1-point BIDQ increase in IBD patients was associated with a 4.6-fold increase in depression (*p* = 0.025), after adjusting for clinical and demographic factors. **Conclusions**: Body image dissatisfaction is prevalent in IBD patients and may be a common feature across chronic diseases. Body image dissatisfaction strongly associates with depression, highlighting the importance of addressing it in IBD management.

## 1. Introduction

Inflammatory bowel disease (IBD) is a chronic, relapsing immune gut disease with complex aetiology, thought to occur due to the interaction of environmental factors and genetic susceptibility [1]. There is a wide spectrum of disease severity, and it can lead to marked morbidity [2]. Treatment with immunomodulatory and suppressive therapies can induce remission and prevent morbidity, but entail a burden of side effects and complications [3]. Surgical procedures are necessary to treat the more severe phenotypes of IBD but may also leave scars, altered bowel function, and stomas [4]. Epidemiological studies suggest that IBD is increasing in incidence worldwide, with associated compound prevalence and pressure on health care systems worldwide. The age-specific incidence rate of IBD in Canterbury, New Zealand (NZ), saw a 1.6-fold increase in the ten years from 2004 to 2014, to 39.5 per 100,000 of population [5]. A recent update in 2023 showed incidence stabilising to 30.1 per 100,000. However, the prevalence has more than doubled since 2005, from 308.3 to 671.4 per 100,000 [6]. Worldwide, the estimated age-standardised prevalence rate for IBD has increased from 79.5 to 84.3 per 100,000 population from 1990 to 2017 [7].

Body image is a concept that refers to a person’s cognitive disposition towards their physical self. It denotes the constellation of thoughts and beliefs which contribute to the person’s idea of their embodied self. Disease and treatment alter the body and the person’s sense of its integrity and resilience, of its fitness to function and to deliver on its capabilities [8]. Body image dissatisfaction refers to a person’s negative appraisal of their body. This may be influenced by cultural norms, socialization, interpersonal interactions, events, and the person’s own personality, thoughts, and coping strategies [9]. Body image dissatisfaction is known to affect patients with IBD. The extent to which depends on disease-related factors as well as other demographic features and personal factors. In a self-reported survey of patients with IBD in Adelaide, South Australia, 66.8% of responders reported that having IBD impaired their body image. Female gender and operative treatment were found to be significant risk factors for a negative view of body image [10].

In a prospective study of IBD patients in Rhode Island, USA, Saha et al. found that body image dissatisfaction in patients with IBD was unchanged over time, despite improvements in the level of disease activity. Again, body image dissatisfaction was greater in women; those with a higher burden of symptoms; those with ileocolonic Crohn’s compared to those with colonic Crohn’s; those with extra-intestinal manifestations of IBD (particularly musculoskeletal and dermatologic); and with steroid exposure, particularly with longer duration [11]. Existing research strongly supports the view that body image dissatisfaction is a major concern among IBD patients. Importantly, an amplified sense of body image dissatisfaction firmly correlates with a reduced quality of life. Expanding upon this understanding, McDermott et al. used two separate, validated, body image assessment questionnaires to study body image dissatisfaction in patients with IBD. Female gender, younger age, and smoking were found to be associated with higher body image dissatisfaction scores. Higher scores were also seen with steroid use, surgery, and active disease. Body image dissatisfaction was associated with low general and IBD-specific quality of life, low self-esteem, low sexual satisfaction, and high levels of anxiety and depression [12]. In a Swiss survey of IBD patients, a statistically significant correlation was found between recurrence of IBD and patient-reported symptoms of anxiety and depression [13].

Indeed, these findings concerning body image disturbance in IBD patients have been corroborated by the first systematic review on this subject [14]. The review, which analysed data from 57 studies, found that body image dissatisfaction is prevalent in IBD patients, particularly among certain demographic groups—such as younger age, female gender, and higher BMI—in some studies, as well as in those with specific disease characteristics like active disease status and steroid use. Moreover, it reinforced the strong link between heightened body image dissatisfaction and decreased quality of life. Despite these significant findings, the review also underscored the need for future research using validated tools due to methodological inconsistencies and heterogeneity in the existing studies.

Thus, it is clear from the literature that body image dissatisfaction and IBD interact to affect psychological well-being. What is not yet clear is if the impact of body image dissatisfaction in IBD is more severe, less severe, or comparable to that seen in other chronic diseases, or whether it is a common effect observed across chronic conditions. Our study was designed to address this knowledge gap. We chose to use the Body Image Disturbance Questionnaire (BIDQ), a measure less commonly deployed in the IBD population [15], alongside the RAND 36 measure [16], which permitted a broad evaluation of disease impact on quality of life. We elected to place IBD in direct comparison with another chronic and multi-system condition—Type 1 Diabetes Mellitus (T1DM)—and healthy, age-matched controls. Other inflammatory conditions, such as rheumatoid arthritis (RA) and inflammatory skin diseases, were considered. However, RA typically presents later in life, introducing potential confounding due to age-related psychosocial differences, while inflammatory skin conditions were excluded because visual changes directly impact body image. T1DM provides a suitable benchmark, given its early onset and enduring nature, akin to IBD. It also shares the challenges of chronic disease management without the gastrointestinal symptoms, surgeries, or visible scars that could disproportionately influence body image dissatisfaction.

## 2. Materials and Methods

Patients who visited consecutively, in person, to the IBD clinic at Hutt Valley Hospital, New Zealand, were invited to participate in the study. We included all adults over 16 years of age with a previous diagnosis of IBD of more than six months’ duration based on clinical, endoscopic, and histological findings. Those undergoing surgery within the last four weeks, who were pregnant, or had other significant chronic medical comorbidities (including diabetes mellitus) were excluded from the study.

The two comparison groups were recruited at the same time: a control group of patients with T1DM, and a control group of healthy volunteers.

T1DM patients over 16 years of age attending the diabetes clinic at Hutt Valley Hospital in person were included. Individuals with T1DM were not included in the study if they had another chronic gastrointestinal condition or were expecting a child. Healthy controls were recruited through noticeboard and newsletter advertising, and word of mouth at Hutt Valley Hospital. Participants in the control group were excluded if they had a diagnosis of a chronic illness or were pregnant.

Age and gender matching were performed across the three groups upon completion of recruitment. Each IBD patient was matched by gender and within a 10-year age band with corresponding participants in both the T1DM and healthy control groups. When multiple matching candidates were available within a band, the subject closest in age to the IBD patient was selected.

To maintain the integrity of the comparative approach, any unmatched IBD patients, or those who did not have a corresponding participant in either the T1DM or healthy control groups, were excluded from further analysis. Correspondingly, any unmatched T1DM or healthy control participants were also removed from the final analysis.

Informed consent was obtained in writing before participants entered the study. The study adhered to applicable New Zealand laws and the Health Information Privacy Code 1994. Ethical approval was granted by the University of Otago Human Ethics Committee (reference number: H16/108).

Demographic information was collected from all participants, including age, gender, ethnicity, co-morbidities, medication use, smoking status, relationship status, and body mass index. Additional data related to their chronic condition was gathered for participants with IBD and T1DM.

All participants completed three questionnaires. To assess body image dissatisfaction, the Body Image Disturbance Questionnaire (BIDQ) (Figure 1), a validated seven-item tool, was utilized. This questionnaire enables individuals to evaluate and rate various aspects of their body image on a continuous scale [15]. The outcome is expressed as the average score derived from the seven individual questions, referred to as the BIDQ score.

All participants completed the Hospital Anxiety and Depression Scale (HADS), a 14-item psychological screening instrument. This tool has been validated for use in comparing clinical groups and shows strong correlations with disease progression, treatment outcomes, and quality-of-life metrics [17,18]. Participants were considered to exhibit anxiety or depression if their score was >7 for the relevant measure.

Participants also completed the RAND 36 Health-Related Quality of Life questionnaire, a widely utilized and thoroughly validated tool that assesses quality of life across multiple domains, including physical functioning, limitations due to physical health, bodily pain, social functioning, mental health (psychological distress and well-being), emotional role limitations, energy levels, and overall health perceptions [16].

All IBD patients had clinical activity of disease calculated at the time of assessment. In Crohn’s disease, the Harvey Bradshaw Index (HBI) [19] was used for the majority of cases; for one patient, the Crohn’s Disease Activity Index (CDAI) [20] was used. Patients were considered to be in clinical remission if the HBI was less than 5 or the CDAI was less than 150 for Crohn’s disease, and if the Simple Clinical Colitis Activity Index (SCCAI) [21] was 2 or less for ulcerative colitis [22].

Each participant was assigned a unique study number, and all completed questionnaires were marked solely with this number, ensuring no other identifying information was included.

### Statistical Methods

Data were analyzed using SPSS Statistics for Windows, Version 28.0 (IBM SPSS Statistics, Armonk, NY, USA: IBM Corp). The primary analysis involved comparing BIDQ scores across the three groups using one-way analysis of variance with Bonferonni’s correction. Secondary outcome measures, including QoL measures and HADS scores, were compared using Student’s *t*-tests. Pearson’s correlation was used to assess relationships between continuous variables, such as age and BIDQ scores. Logistic regression was performed on all included IBD patients to investigate the association of BIDQ with demographics and measures of well-being.

Previous research indicated that the BIDQ mean score for healthy individuals was 1.57 (SD 0.6, *n* = 53) in men and 1.81 (SD 0.67, *n* = 198) in women [23]. To estimate sample size, we used those figures, assuming a sample that was 50% male and a clinically significant difference of 0.5. While no definition of clinical significance for BIDQ exists in the literature, McDermott et al. [12] did show a difference of 0.6 in their patients with IBD compared to normative values. Thus, we chose 0.5 as a reasonable difference that might be expected when comparing IBD to healthy control subjects. This gave a minimum sample size of 37 in each group, with a power of 90% and significance of 0.05 (with Bonferroni’s correction).

## 3. Results

T A total of 161 subjects were recruited, comprising 50 IBD patients, 55 healthy control subjects, and 56 T1DM patients. The study included 45 matched pairs by age and gender to compare IBD patients with healthy controls, and 38 matched pairs to compare IBD patients with T1DM controls. One IBD patient, 18 T1DM patients, and 10 control subjects were excluded because there was no available matched subject. A total of 132 subjects were included in the final analysis of results (Figure 2).

Participant baseline characteristics are outlined in Table 1. IBD patient background and disease activity are outlined in Table 2.

Mean BIDQ was compared between IBD patients and matched healthy controls, and IBD patients and matched T1DM patients. The mean BIDQ was statistically significantly higher in IBD patients than healthy controls (2.05 vs. 1.58, *p* = 0.004). No significant difference was seen between matched IBD patients and T1DM patients (2.03 vs. 1.72, *p* = 0.09) (Figure 3).

When comparing BIDQ scores in IBD patients with active disease to those in clinical remission (defined as HBI < 5, CDAI < 150, or SCCAI ≤ 2), patients with inactive disease showed a tendency toward lower BIDQ scores; however, this difference did not reach statistical significance (mean BIDQ 1.93 vs. 2.36, *p* = 0.08). Additionally, no significant relationship was observed between age and BIDQ score among IBD patients (r = −0.123, *p* = 0.398). Similarly, no significant difference was seen in BIDQ between males and females (2.04 vs. 2.05, *p* = 0.99).

There was a statistically significant difference in the mean depression score of the HADS between matched IBD patients and healthy controls (6.51 vs. 3.87, *p* = 0.002). There was no statistically significant difference observed between IBD patients and their T1DM counterparts (6.5 vs. 6.81, *p* = 0.8).

Additionally, no significant difference was seen in the anxiety score of the HADS between IBD and healthy controls (5.51 vs. 4.89, *p* = 0.44) or IBD and T1DM patients (5.28 vs. 4.27, *p* = 0.24).

The RAND SF-36 quality of life measures were compared between IBD patients and healthy controls. The IBD patients were found to have significantly lower (worse) subscores than controls for role limitations due to physical health (60.8 vs. 87.8, *p* < 0.001), energy and fatigue (42.5 vs. 56.8, *p* = 0.005), emotional well-being (64.5 vs. 73.1, *p* = 0.042), social functioning (71.9 vs. 84.2, *p* = 0.019), pain (68.1 vs. 82.4, *p* = 0.003), and general health (36.2 vs. 59.4, *p* < 0.001). However, no significant differences were seen for physical functioning (82.9 vs. 89.4, *p* = 0.098) or role limitations due to emotional problems (56.1 vs. 71.9, *p* = 0.09).

Conversely, when the same analysis was performed comparing IBD patients with T1DM patients, the only significant difference was observed in the mean score for energy and fatigue in IBD vs. T1DM patients (45.4 vs. 57.1, *p* = 0.046). No significant differences were seen in the other RAND SF-36 subscores (data presented in Figure 4).

Given the elevated levels of depression in IBD patients, a logistic regression was performed to ascertain the effects of age, BIDQ, BMI, smoking status, gender, clinical remission, and relationship status on the likelihood of depression (HADS depression score > 7) in all subjects with IBD. The overall model was evaluated using the Likelihood Ratio Test and was found to be statistically significant, X^2^ (8 df)= 20.75, *p* = 0.008. The model explained 51.1% (Nagelkerke R^2^) of the variance in depression and correctly classified 79.5% of cases. Increased BIDQ score was significantly associated with the likelihood of depression. For every one-point increase in BIDQ, the likelihood of depression increased 4.2 times (OR 4.2, 95%CI [1.08,16.24]). Age, BMI, smoking status, gender, and relationship status were not significantly associated with depression. Because no similar association between HADS anxiety score and BIDQ was seen, regression analysis was not performed for the anxiety score.

## 4. Discussion

In this case-control study, we sought to examine the complex relationship between body image dissatisfaction and IBD. We compared this to body image dissatisfaction in healthy individuals and those with another chronic disease, T1DM. Our analysis showed a significantly increased rate of body image dissatisfaction in IBD patients when compared to healthy controls. This is similar to the effect seen in other studies examining body image in IBD [10,11,12], including a meta-analysis of existing literature [14]. Conversely, we found a comparable body image dissatisfaction burden in matched T1DM patients, in keeping with previous studies that have demonstrated raised levels of body image dissatisfaction in T1DM [25,26,27,28]. Previous studies have demonstrated increased levels of body image dissatisfaction in IBD patients when compared to normative values; however, none have compared body image dissatisfaction in IBD patients compared to normal controls. This is important as the demographics of IBD patients enrolled into studies may vary from the demographics seen in the original studies that defined normative values for the scores.

An association between body image dissatisfaction and psychological health, including depression and anxiety, has been seen in numerous disease populations previously [29]. Meta-analyses have shown that body appreciation is negatively associated with psychological well-being [30]. Studies of breast cancer [31], postpartum depression [32], and systemic lupus erythematosus [33] have all shown similar outcomes.

While body image dissatisfaction levels across IBD and T1DM patients did not differ significantly, there were increased levels of depression seen in IBD patients. Logistic regression analysis indicated an association between body image dissatisfaction and increased levels of depression in the IBD cohort. Conversely, conventional demographic parameters, such as gender, age, and lifestyle practices like smoking, did not appear to have a significant effect. This suggests a potential critical role of body image dissatisfaction in the mental health outcomes of IBD patients.

Our study highlights the significant impact of body image dissatisfaction on the quality of life in individuals with inflammatory bowel disease (IBD). Prior research has shown that body image dissatisfaction correlates with reduced quality of life in IBD patients, affecting physical, emotional, and social domains [12].

We found no significant association between clinical remission and body image dissatisfaction scores, nor did we see significant effects of age or gender on body image dissatisfaction scores in IBD patients. This contrasts with previous studies and may suggest that the impact of clinical remission, age, and gender on body image dissatisfaction in IBD patients could vary based on specific population characteristics, sample size, or cultural factors influencing body image perception. Alternatively, it could indicate that body image dissatisfaction is a persistent issue for IBD patients, less influenced by disease activity or demographic variables than previously thought. Further studies with larger and more diverse samples are needed to clarify these relationships and explore other factors that may drive body image dissatisfaction in this population

Regarding treatment options for body image dissatisfaction, there is limited research specifically addressing this issue in IBD patients. In other settings, meta-analyses have shown that interventions can have positive but small effects [34]. While these interventions have not been extensively studied in IBD, the potential exists for adapting similar approaches to address body image dissatisfaction in this population, especially given the mental health challenges frequently reported in IBD patients.

Our study is not without limitations. Although the sample size is statistically sufficient, its relatively small scale may limit the generalizability of the findings. Additionally, the study’s cross-sectional design allows only for the inference of associations and not causative links. This is a single-center study, and our participants were predominately NZ Europeans, which may limit the applicability of the findings to other contexts. Furthermore, the reliance on self-reported data inherently carries a risk of bias in participants’ responses. There is a pressing need for future longitudinal studies with more substantial sample sizes to further validate and elaborate on these findings. However, there are strengths to our study design. This is one of few studies to use control participants to compare outcome measures, rather than normative values, and the only study to include a disease control group. Additionally, matching each of our controls by age and gender with IBD patients enabled us to more reliably attribute observed differences in outcomes to disease presence rather than age or gender discrepancies.

## 5. Conclusions

In summary, this study emphasizes the often-overlooked issue of body image dissatisfaction within the IBD context, showing its significant impact on patients’ psychological health. It has added to the existing body of evidence that body image dissatisfaction is significant in chronic disease, including inflammatory bowel disease. It has also highlighted significant need for psychological and mental health input to identify and support those with body image dissatisfaction amongst IBD patients. Although specific intervention and timing of psychological input have not been investigated in this study, it nonetheless demonstrates a need for clinical resources and further research in this area.

## Figures and Tables

**Figure 1 nutrients-17-00015-f001:**
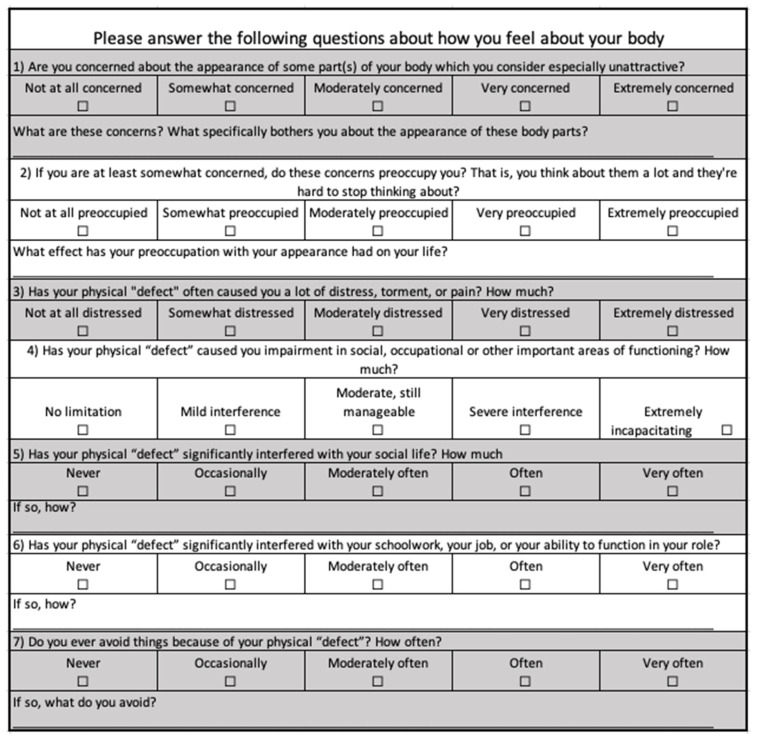
The Body Image Disturbance Questionnaire [15].

**Figure 2 nutrients-17-00015-f002:**
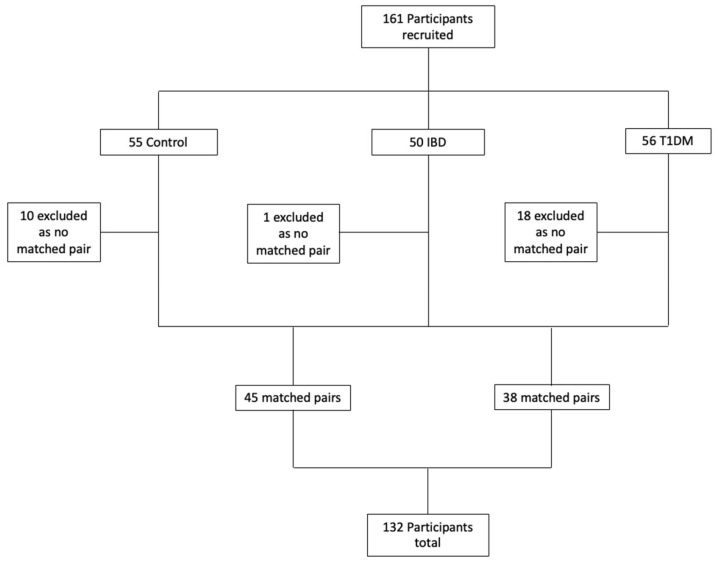
Participant Flow Diagram.

**Figure 3 nutrients-17-00015-f003:**
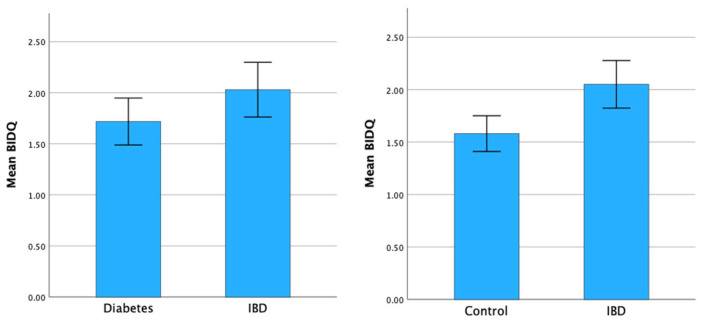
Mean BIDQ compared between groups with 95% confidence intervals.

**Figure 4 nutrients-17-00015-f004:**
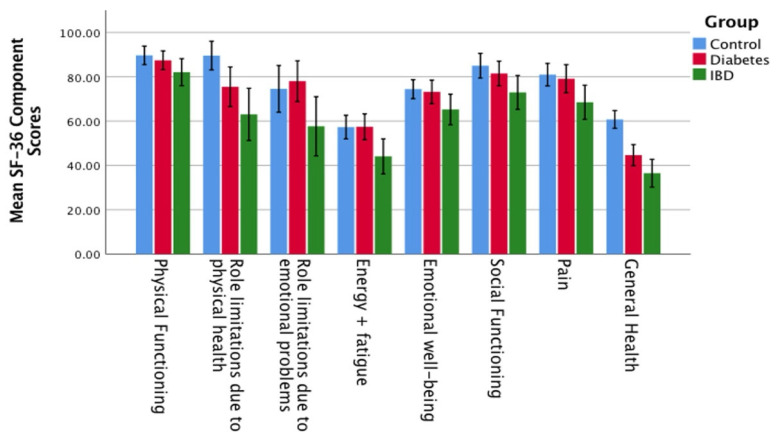
RAND SF-36 component scores with 95% confidence intervals.

**Table 1 nutrients-17-00015-t001:** Participant demographics.

	IBD Patients	T1DM Patients	Control Subjects
Mean age in years (s.d.)	40.2 (12.4)	36.7 (13.1)	39.0 (12.7)
Gender (% female)	63.3%	52.6%	68.9%
Mean BMI (s.d.)	28.2 (6.5)	27.4 (4.6)	25.9 (4.6)
Smoking status			
Current (*n*, %)	9 (18.4%)	0	4 (8.9%)
Ex (>28 days) (*n*, %)	7 (14.3%)	5 (13.2%)	10 (22.2%)
Never (*n*, %)	31 (63.3%)	33 (86.8%)	31 (68.9%)
Unknown (*n*, %)	2 (4.1%)	0	0
Relationship status			
De facto or married (*n*, %)	28 (57.1%)	26 (68.4%)	29 (64.4%)
Single (*n*, %)	12 (24.5%)	10 (26.3%)	15 (33.3%)
Unknown (*n*, %)	9 (18.4%)	2 (5.3%)	1 (2.2%)
Ethnicity			
NZ European (*n*, %)	47 (95.9%)	35 (92.1%)	20 (44.4%)
Māori (*n*, %)	0	1 (2.6%)	3 (6.7%)
Pacific Peoples (*n*, %)	1 (2.0%)	1 (2.6%)	1 (2.2%)
Asian (*n*, %)	1 (2.0%)	1 (2.6%)	19 (42.2%)
Other Ethnicity (*n*, %)	0	0	2 (4.4%)

s.d. = standard deviation.

**Table 2 nutrients-17-00015-t002:** Description of IBD phenotype.

	IBD Patients
Median age at diagnosis in years (IQR)	26 (20–37)
Median IBD duration in years (IQR)	9 (3–18.5)
IBD type	
• Ulcerative colitis (N)	13
◦ E1—proctitis	3
◦ E2—left-sided	7
◦ E3—pancolitis	3
• Crohn’s disease (N)	33
Distribution	
◦ L1—ileal	9
◦ L2—colonic	7
◦ L3—ileocolonic	17
Behaviour	
◦ B1—non-stricturing, non-fistulising	11
◦ B2—stricturing	10
◦ B3—fistulising	12
◦ P—perianal	9
• IBD Unclassified (N)	3
Steroid use in last 12 months (N, %)	10 (20%)
Clinical remission (N, %)	36 (74%)
Past abdominal surgery (N, %)	12 (25%)

Description of IBD phenotype is according to the Montreal classification [24], steroid use, and disease activity in IBD group. IQR = interquartile range. Clinical remission = HBI less than 5, CDAI less than 150, or SCCAI of 2 or less.

## Data Availability

The data presented in this study are available on request from the corresponding author due to privacy restrictions.

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
