# Peer review of "Body Image Dissatisfaction Is Increased in Inflammatory Bowel Disease Compared to Healthy Matched Controls but Not Diseased Controls: A Case-Control Study from New Zealand"

_nutrients, 2024, doi:10.3390/nu17010015_

Round 1

Reviewer 1 Report

Comments and Suggestions for Authors

1.

  1. It is hard to follow manuscript when you have two similar abbreviations, IBD and BDI, please use full terms 
  2. Line 1 - type of paper 
  3. Add study setting to the title and abstract 
  4. Check whole manuscript for double spaces, also before references space should be used 
  5. Introduction could be improved, flow is missing. Have larger paragraphs please, with same theme. 
  6. Line 52 - reference is missing 
  7. Same for line 73
  8. Why is one sentence a paragraph, as in line 80
  9. Please delete mistakenly included text lines 106-114
  10. Improve quality and add reference for figure 1 
  11. Improve quality of figure 2
  12. Table 2 can be added as supplement and just described in manuscript 
  13. Discussion section must be improved, compare your findings with relevant studies from last 5 years 
  14. English language and scientific flow could be improved throughout the whole manuscript 

This is interesting study but I am not sure how it adds to the existing body of literature on this matter. Maybe if the manuscript gets improved it will be more relevant in the field.

Author Response

We would first like to thank the reviewer for their positive comments and their careful review and suggestions. Please find below our response to the reviewer’s comments and the changes made to the resubmitted manuscript.

Reviewer 1

Reviewer comment 1. It is hard to follow manuscript when you have two similar abbreviations, IBD and BDI, please use full terms

Response 1. We thank the reviewer for this comment, something we have also considered. BID is the commonly accepted abbreviation in the literature and has been used in other papers regarding IBD, however we agree with the reviewer’s comment and have elected to keep the acronym IBD but write “body image dissatisfaction” in full throughout the paper except where it is used as part of the “BIDQ score”.

Reviewer comment 2. Line 1 - type of paper

Response 2. Thank you for picking up on this oversight. It has been corrected.

Reviewer comment 3. Add study setting to the title and abstract

Response 3. We have added the geographical context to the title and abstract as well as expanding on the clinical context in the abstract.

New title: “Body image dissatisfaction is increased in inflammatory bowel disease compared to healthy matched controls but not diseased controls: A case-control study from New Zealand.”

Additional text line 13: Methods: In this New Zealand based case-control study conducted in a secondary care hospital,

Reviewer comment 4. Check whole manuscript for double spaces, also before references space should be used

Response 4. All double spaces have been removed. Thank you for noticing this.

We have also ensured there are spaces before each reference.

Reviewer comment 5. Introduction could be improved, flow is missing. Have larger paragraphs please, with same theme.

Response 5. Thank you for your suggestions. We have adopted the approach of having 4 main sections to the introduction: 1. Introduction to IBD 2. Introduction to body image dissatisfaction 3. Literature regarding body image and IBD 4. Discussion of body image in the setting of chronic disease and justification of study design choices (as suggested by the second reviewer). We have joined smaller paragraphs to form larger blocks of text that follow this structure, as suggested. We hope this approach ensures adequate flow and theming.

Reviewer comment 6. Line 52 - reference is missing

Same for line 73

Response 6. Line 52 refers to the same reference as the sentence above. The reference has been moved down rather than citing twice in a row. The whole of the paragraph that includes line 73 refers to the McDermott study. This is cited at the end of the paragraph.

Reviewer comment 7. Why is one sentence a paragraph, as in line 80

Please delete mistakenly included text lines 106-114

Improve quality and add reference for figure 1

Improve quality of figure 2

Response 7. All completed.

Reviewer comment 8. Table 2 can be added as supplement and just described in manuscript

Response 8. Given that reviewer 2 asked for more information about the IBD phenotype of the cohort and required further comment on this we are not able to respond to this request. Along with reviewer 2 we believe that the nature of the IBD that the participants experience, and the activity and complications of it, are relevant to the applicability of the study and would prefer to leave table 2 as it stands.

Reviewer comment 9. Discussion section must be improved, compare your findings with relevant studies from last 5 years

Response 9. Thank you for the suggestion to strengthen our discussion by providing greater comparison with previous relevant studies. We have gone about doing so, in particular we have made the following changes which we hope will adequately answer the reviewer’s request:

Line 287, addition of literature regarding previous studies of body image dissatisfaction in IBD and T1DM compared to our results.

Line 298 addition of a section regarding findings of similar studies in other chronic diseases.

The recommendation to update our literature review looking for studies in the last 5 years was gratefully accepted. We conducted a pubmed search looking and the search terms “body image” and (“inflammatory bowel disease” or Crohn* or IBD or “ulcerative colitis). This search revealed 271 related records, all of which were reviewed and only 5 new related articles found. Four of those referred specifically to a paediatric population and were not relevant to our discussion. The one adult article was a study of eating disordered behaviour specifically and not relevant to our study. For this reason we were only able to strengthen our discussion by referring to the T1DM literature and drawing conclusions from literature already outlined in our introduction. We trust that the reviewer will find the changes we have made sufficiently strengthen our discussion.

Reviewer comment 10. English language and scientific flow could be improved throughout the whole manuscript

Response 10. We thank the reviewer for their input but would like to note that the majority of the authors are senior gastroenterologists, all were trained in NZ English language universities and the senior author has a research degree from a London UK university. We have all read the manuscript and are confident in the accuracy of the English language used.

Reviewer comment 11. This is interesting study but I am not sure how it adds to the existing body of literature on this matter. Maybe if the manuscript gets improved it will be more relevant in the field.

Response 11. We thank the reviewer for their interest in our study. We believe our article adds to the body of literature by taking the novel approach of comparing two distinct chronic diseases, which has never been done before. While there are existing studies of body image dissatisfaction in IBD many have methodological inconsistencies and few use healthy controls for comparison, rather than normative values. Normative values have been shown to vary between cultural contexts, as pointed out by reviewer 2, and using a control cohort which is age and gender matched to the disease cohorts of interest is a strength of our study and an addition to the existing literature.

Reviewer 2 Report

Comments and Suggestions for Authors

The study by Inns et al. investigates body image dissatisfaction (BID) among patients with inflammatory bowel disease (IBD), comparing it to healthy controls and individuals with type 1 diabetes mellitus (T1DM). Using validated questionnaires, the authors found that IBD patients had significantly higher BID scores than healthy controls but similar scores to T1DM patients. Depression levels were also higher among IBD patients, with a strong association between BID and depression, independent of demographic factors or disease remission status. The study highlights the psychological impact of BID on IBD patients, suggesting it may be a characteristic of chronic diseases in general. This is a beautifully done study with valuable insights. However, the authors acknowledge some limitations such as modest sample size. I have some comment to the authors:

1. The study is conducted in a single center in New Zealand, with a predominantly NZ European cohort. Discuss how cultural perceptions of body image might limit the applicability of the results to more diverse populations.

2.  The manuscript reports a statistically significant difference in BID scores between IBD patients and healthy controls (2.05 vs. 1.58, p=0.004). However, given that this difference is only 0.47 on a 1-to-5 scale, its clinical relevance appears limited. The authors should address whether such a small difference is meaningful in the context of patient care or psychological burden. Additionally, consideration of established thresholds for clinically significant BIDQ score changes, if available, could help contextualize these results.

3. The manuscript lacks detailed clinical information on the IBD cohort, which limits the ability to interpret the findings in the context of disease severity and management. Specifically, data on inflammatory markers such as CRP and fecal calprotectin, disease activity indices like CDAI or Mayo score, and prior or current treatment regimens, including the use of biologics, are not reported. These variables are critical for understanding how disease activity and treatment may influence body image dissatisfaction (BID). Including such information would provide a more comprehensive analysis of potential confounding factors and strengthen the study's clinical relevance.

4. The flow diagram (Figure 2) and descriptive statistics tables (e.g., Table 1) are useful, but including visual representations of key findings, such as BIDQ scores across groups, could enhance reader engagement and clarity.

5. While the use of T1DM as a comparator strengthens the study’s design, the rationale for this choice is not fully justified, as T1DM and IBD may differ significantly in psychosocial impact. T1DM primarily involves self-management of glucose levels and dietary restrictions, which may not directly align with the physical and psychosocial burdens of IBD, such as surgery, visible scars, and gastrointestinal symptoms. A more appropriate comparator might be rheumatoid arthritis (RA). Both IBD and RA share features such as: use of immunosuppressive therapies and steroids, with similar side effects like weight gain, acne, and mood disturbances. In addition, in RA there is an impact on physical function, social activities, and overall quality of life, closely mirroring the challenges faced by IBD patients. If T1DM is retained as a comparator, the authors should explicitly justify this choice by discussing its similarities and differences with IBD regarding psychosocial and physical health challenges.

Author Response

We would first like to thank the reviewer for their positive comments and their careful review and suggestions. Please find below our response to the reviewer’s comments and the changes made to the resubmitted manuscript.

Reviewer 2

Reviewer comment 1. The study is conducted in a single center in New Zealand, with a predominantly NZ European cohort. Discuss how cultural perceptions of body image might limit the applicability of the results to more diverse populations.

Response 1. We have additionally acknowledged this limitation with the following text at line 312: This is a single centre study and our participants were predominately NZ Europeans, which may limits the applicability of the findings to other contexts.

Reviewer comment 2. The manuscript reports a statistically significant difference in BID scores between IBD patients and healthy controls (2.05 vs. 1.58, p=0.004). However, given that this difference is only 0.47 on a 1-to-5 scale, its clinical relevance appears limited. The authors should address whether such a small difference is meaningful in the context of patient care or psychological burden. Additionally, consideration of established thresholds for clinically significant BIDQ score changes, if available, could help contextualize these results.

Response 2. In our statistical plan we considered a clinically significant difference in the BIDQ to be 0.5 (line 189). This was based on previous research by McDermott et al. [1] which showed that “the mean score (with a possible scoring range from 0–4) was 2.2 for men and 2.4 for women. Although we had no healthy control group available for comparison, historical data indicate mean scores of 1.6 for healthy men and 1.8 for healthy women.” Based on these findings and the same normative values we concluded that we could expect a difference of around 0.5 and chose that as the clinically significant difference for our sample size calculation. While differences as high as 1.4 are seen when patients with anorexia nervosa are compared to healthy controls [2], we did not expect anywhere near that effect size for IBD or T1DM. As such, while not definition of a clinically significant difference for BIDQ exists in the literature to our knowledge, we considered a difference of 0.5 to be demonstrably clinically important. To strengthen the description regarding this assumption we have added the following text at line 190: “While no definition of clinical significance for BIDQ exists in the literature, McDermott et al. [12] did show a difference of 0.6 in their patients with IBD compared to normative values, thus we chose 0.5 as a reasonable difference that might be expected when comparing IBD to healthy control subjects.”

Reviewer comment 3. The manuscript lacks detailed clinical information on the IBD cohort, which limits the ability to interpret the findings in the context of disease severity and management. Specifically, data on inflammatory markers such as CRP and fecal calprotectin, disease activity indices like CDAI or Mayo score, and prior or current treatment regimens, including the use of biologics, are not reported. These variables are critical for understanding how disease activity and treatment may influence body image dissatisfaction (BID). Including such information would provide a more comprehensive analysis of potential confounding factors and strengthen the study's clinical relevance.

Response 3. We agree with the reviewer that the nature and activity of the IBD participant’s disease is important to understanding the group’s make up. Table 2 attempts to describe the nature of the IBD extent and location as well as complications using the Montreal classification. In addition we prospectively collected validated clinical scores on participants at the time of the study measurements. We did not require patients to complete a contemporaneous CRP as we did not consider that to be a useful measure of activity at a single point in time. Faecal calprotectin may have been a more useful tool but we were not in a position to require this of all participants and were concerned that it would be difficult to maintain a contemporaneous association between sample collection and the measures taken in the study. We are not in a position to provide further biochemical markers of disease activity but hope the reviewer will agree that the inclusion of  clinical activity scores goes some way to addressing this concern.

Reviewer comment 4. The flow diagram (Figure 2) and descriptive statistics tables (e.g., Table 1) are useful, but including visual representations of key findings, such as BIDQ scores across groups, could enhance reader engagement and clarity.

Response 4. We thank the reviewer for this comment. We had considered including this figure but were concerned it was repetitive considering the data was well described in the text. We have now included this figure at line 226 and updated the figure numbers and references to them in the text.

Reviewer comment 5. While the use of T1DM as a comparator strengthens the study’s design, the rationale for this choice is not fully justified, as T1DM and IBD may differ significantly in psychosocial impact. T1DM primarily involves self-management of glucose levels and dietary restrictions, which may not directly align with the physical and psychosocial burdens of IBD, such as surgery, visible scars, and gastrointestinal symptoms. A more appropriate comparator might be rheumatoid arthritis (RA). Both IBD and RA share features such as: use of immunosuppressive therapies and steroids, with similar side effects like weight gain, acne, and mood disturbances. In addition, in RA there is an impact on physical function, social activities, and overall quality of life, closely mirroring the challenges faced by IBD patients. If T1DM is retained as a comparator, the authors should explicitly justify this choice by discussing its similarities and differences with IBD regarding psychosocial and physical health challenges.

Response 5. We thank the reviewer for this comment and acknowledge the importance of justifying the choice of T1DM as a comparator.

While other inflammatory conditions, such as rheumatoid arthritis (RA) and inflammatory skin diseases, were considered, they were excluded for specific reasons. RA typically presents later in life, introducing potential confounding due to age-related psychosocial differences, whereas inflammatory skin conditions have a direct visual impact on body image, which could disproportionately influence body image dissatisfaction (BID) scores.

We retained T1DM as the comparator because it provides a suitable benchmark, given its early onset and enduring nature, which are similar to IBD. T1DM also shares the challenges of long-term disease management, including dietary restrictions and self-monitoring, without the confounding factors of gastrointestinal symptoms, surgeries, or visible scars often seen in IBD. This allows for a clearer assessment of the impact of chronic disease on body image dissatisfaction.

In response, we have added a section at line 100 to clarify our rationale for selecting T1DM.We believe this addition provides a more robust justification for our choice and addresses the reviewer’s concerns.

Round 2

Reviewer 1 Report

Comments and Suggestions for Authors

Acceptable for publication